# Histone Modifications Represent a Key Epigenetic Feature of Epithelial-to-Mesenchyme Transition in Pancreatic Cancer

**DOI:** 10.3390/ijms24054820

**Published:** 2023-03-02

**Authors:** Ying Xu, Qing Zhu

**Affiliations:** Abdominal Oncology Ward, Cancer Center, West China Hospital, Sichuan University, Chengdu 610041, China

**Keywords:** pancreatic cancer, epigenetics, histone modification, epithelial-to-mesenchymal transition, pancreatic ductal adenocarcinoma

## Abstract

Pancreatic cancer is one of the most lethal malignant diseases due to its high invasiveness, early metastatic properties, rapid disease progression, and typically late diagnosis. Notably, the capacity for pancreatic cancer cells to undergo epithelial–mesenchymal transition (EMT) is key to their tumorigenic and metastatic potential, and is a feature that can explain the therapeutic resistance of such cancers to treatment. Epigenetic modifications are a central molecular feature of EMT, for which histone modifications are most prevalent. The modification of histones is a dynamic process typically carried out by pairs of reverse catalytic enzymes, and the functions of these enzymes are increasingly relevant to our improved understanding of cancer. In this review, we discuss the mechanisms through which histone-modifying enzymes regulate EMT in pancreatic cancer.

## 1. Introduction

Pancreatic cancer is the seventh leading cause of cancer-related death globally [1]. Patient prognosis is bleak, with a five-year survival rate of only 10% [2,3]. Moreover, the incidence of pancreatic cancer has risen steadily in recent years [4]. By 2025, it is expected to become the second-leading cause of cancer-related mortality in the US [1]. Surgery remains the main therapeutic approach but is only effective during the early stages of disease. The early metastases and rapid progression commonly seen in patients result in diagnosis not occurring until the advanced stages of disease in more than half of all cases, which precludes them from surgery [5,6]. Although systemic combination chemotherapy also forms a crucial component of pancreatic cancer treatment, drug resistance is a significant problem [7]. Therefore, it is essential to understand the molecular mechanisms involved in the carcinogenesis, progression, metastasis, and drug resistance of pancreatic cancer to identify precise diagnostic and therapeutic targets.

Epithelial–mesenchymal transition (EMT) is a fundamental process for embryogenesis, organ development, and tissue repair, as well as for tumor invasion and early metastasis [8]. During EMT, epithelial cells lose their apical–basal polarity and cell–cell adhesion, and are transformed into invasive mesenchymal cells [9]. In addition, EMT can generate cells with stem cell properties that allow them to form tumors more efficiently [10,11]. EMT enables cancer cells at the primary site to acquire motility and invasiveness, which can drive the malignant progression of tumors. Furthermore, EMT also can regulate a variety of cancer features, such as tumor cell stemness, adaptation to microenvironmental alterations, and resistance to therapy [12]. EMT facilitates tumorigenesis and promotes the invasion, metastasis, and therapeutic resistance of pancreatic cancer, and is, therefore, associated with poor patient prognosis [13,14,15].

EMT is triggered by signals sent within the cellular microenvironment. The mechanisms underlying this process are intricate and involve multiple signaling pathways and various EMT-induced transcription factors [16]. Interestingly, the acquisition of mesenchymal properties in cancer cells is transient rather than permanent, and cancer cells can restore their epithelial state through the mesenchymal–epithelial transformation (MET) process [17]. Given the extensive gene expression reprogramming needed to complete the reversible EMT process, epigenetic regulators are likely to have essential functions [18]. Epigenetics describes heritable changes in cellular phenotypes that are independent of alterations in the DNA sequence [19]. Epigenetic changes are associated with aberrant gene functions and altered patterns of gene expression. The major epigenetic regulators include DNA methylation, histone modification, chromatin remodeling, and non-coding RNAs [19,20]. These molecular regulatory mechanisms can alter promoter accessibility and chromatin structure, so as to affect cellular gene expression and influence tissue homeostasis [20]. Indeed, we find that the most significant and complex epigenetic regulatory mechanisms detected within pancreatic cancers comprise histone modifications, which include methylation, acetylation, ubiquitination, phosphorylation, and SUMOylation [21]. Histone modifications can be dynamically regulated by enzymes with reverse catalytic activity, and these enzymes are the primary epigenetic regulators of EMT.

This review provides an overview of EMT in pancreatic cancer, together with the mechanisms of histone modification involved in the regulation of EMT, focusing primarily on methylation, acetylation, and ubiquitination. Furthermore, we discuss the application of histone epigenetic regulators as therapeutics for patients with pancreatic cancer.

## 2. EMT in Pancreatic Cancer

EMT leads to significant phenotypic alterations in cancer cells that are associated with their invasion, spread, and metastasis. Epithelial cells maintain cell–cell contact through tight junctions, adhesive junctions, desmosomes, and gap junctions [22]. During EMT, these cell–cell junctions are lost, and significant cytoskeletal reorganization occurs. Epithelial cells lose top-basal polarity and are transformed into mesenchymal cells, gaining motility and invasiveness [9,23]. The activation of EMT leads to a downregulation of the expression of epithelial genes, including E-cadherin, ZO-1, and occludin. Concurrently, cells acquire a mesenchymal morphology and express mesenchymal markers, such as N-cadherin, vimentin, and fibronectin [8]. The downregulation of E-cadherin (which binds to β-catenin to form mature adhesion junctions between cells [24,25]) is a key step in EMT and is often associated with the invasive potential and undifferentiated phenotype of tumor cells. Moreover, E-cadherin downregulation can disrupt cell adhesions and activate various EMT-related intracellular signaling pathways [26].

EMT is induced by a variety of extracellular stimuli and intracellular signaling pathways, including the transforming growth factor (TGF), Wnt, and hedgehog (Hh) signaling pathways, etc. [16] (Figure 1). TGF-β can induce EMT through different signaling pathways: those that are dependent on the SMAD (suppressor of mothers against decapentaplegic) transcription factor family, and those that are SMAD-independent [27]. In the SMAD-dependent pathway, TGF-β signals through a tetrameric complex containing type I and type II receptors (TβRI and TβRII) to activate SMAD2 and SMAD3. Phosphorylated SMAD2 and SMAD3 form trimers with SMAD4 to regulate EMT-related transcriptional regulators. In the SMAD-independent pathway, TGF-β activates the extracellular signal-regulated kinase 1/2 (ERK)/mitogen-activated protein (MAP) kinase, Rho GTPase, and phosphoinositide 3 (PI3) kinase/Akt pathways to promote EMT [28]. The Wnt signaling pathway is also crucial for the induction of EMT. In the absence of Wnt signaling, β-catenin is phosphorylated by casein Kinase 1 (CK1) and adenomatous polyposis coli (APC)/AXIN/glycogen synthase kinase-3 beta (GSK3β) complexes, prior to being ubiquitinated and degraded by proteasomes. When Wnt ligands bind to frizzled receptors, this degradation process is inhibited, and accumulated cytoplasmic β-catenin transfers to the nucleus and activates the transcription of EMT-related genes [29]. In Hh signaling, Hh ligands bind to the transmembrane receptor patched-1 to inhibit the transmembrane G protein-coupled receptor Smoothened (SMO), initiating intracellular cascades and activating the glioma-associated oncogene (GLI) transcription factors that induce the expression of EMT-related genes [30]. In addition, tyrosine kinase receptors are also involved in pancreatic cancer EMT through the regulation of the PI3K/AKT and ERK/MAPK signaling pathways [31,32]. These signaling pathways activate EMT transcription factors (EMT-TFs) [12], which include the Snail family (Snail1/Snail and Snail2/Slug), TWIST family (Twist1 and Twist2), and Zinc finger E-box binding homeobox family (Zeb1 and Zeb2) of proteins [33].

The tumor microenvironment (TME) is crucial in EMT. The pancreatic cancer microenvironment consists of cancer cells, stromal cells, and extracellular components [34]. TME-inducible factors secreted by cancer cells and cancer-associated fibroblasts (CAF) create an inflammatory microenvironment by recruiting immune cells [35]. Many of these cells secrete cytokines and chemokines that induce EMT [36]. For example, IL-6 regulates the expression of EMT-related genes in pancreatic cancer cells through the STAT3-mediated signaling pathway [37]. As pancreatic cancer contains abundant stromal cells and extracellular matrix but lacks vascularization, severe hypoxia persists within the tumor, causing broad effects on cellular behavior [38]. The adaptive response of pancreatic cancer cells to hypoxia is mainly mediated by hypoxia-inducible factors (HIFs), which bestow more aggressive and therapeutically resistant phenotypes [39]. During hypoxia, the ubiquitin-proteasome degradation pathway mediated by HIF-1 is inhibited. As a consequence, HIF-1α accumulates in cells and binds to HIF-1β to form a functional transcriptional complex [40]. HIF-1α drives hypoxia-induced EMT programming in pancreatic cancer cells through interaction with the nuclear factor kappa-light-chain-enhancer of activated B cells (NF-κB) transcriptional complex [41]. In addition, HIF-2α can promote EMT in pancreatic cancer cells by regulating the binding of Twist2 to the E-cadherin promoter [42].

The two most common genetic alterations in pancreatic cancer are mutations in the Kirsten rat sarcoma viral (KRAS) oncogene and inactivation of SMAD4 [43]. KRAS-driven tumors frequently exhibit EMT induction [15,44,45,46]. The most extensively studied mechanism for inducing EMT is the TGF-signaling pathway [47], and SMAD4 is an effector of this pathway [28,48]. The function of SMAD4 in the EMT of pancreatic cancer cells remains controversial. Whilst the majority of studies have demonstrated that EMT requires the entire TGF-β signaling pathway (including SMAD4) [28,49,50,51], others have shown that the inactivation of SMAD4 can induce EMT [52,53]. Nevertheless, it is clear that SMAD4 is a major factor in the EMT seen in pancreatic cancers.

Cancer stem cells (CSCs) are undifferentiated quiescent cells that possess properties of self-renewal and cellular plasticity [54]. The molecular features of CSCs are influenced by cells exhibiting an EMT phenotype, and CSCs themselves show an EMT phenotype [55,56,57]. Pancreatic cancer stem cells (PCSCs) are subsets of pancreatic cancers with specific cell surface markers that have important functions during tumor recurrence and therapeutic resistance [58,59,60]. Key signaling pathways that regulate PCSCs can also activate many EMT-related pathways, promoting pancreatic cancer progression and resistance to therapeutics [61,62].

## 3. Epigenetics in Pancreatic Cancer

Cancer was initially recognized as a genetic disease. However, it is now widely believed that the initiation and progression of cancer cannot be accounted for by genetic alterations alone, but must also involve epigenetic changes [20]. Epigenetic modifications are reversible changes that regulate gene expression without altering DNA sequences. They include DNA methylation, histone modification, chromatin modification, and alterations in noncoding RNA profiles (Figure 2). Chromatin is the carrier of genetic information, and the nucleosome is its basic unit. Each nucleosome consists of a histone octamer composed of the core histones (two copies each of H2A, H2B, H3, and H4), which is wrapped in approximately 147 bp of DNA [63]. DNA methylation transfers a methyl group to the C5 position of cytosine in CpG dinucleotides to form 5-methylcytosine (5mC). The DNA (cytosine-5)-methyltransferase 3A (DNMT3A) and DNMT3B enzymes are responsible for de novo DNA methylation, which is maintained by DNMT1 [64]. Chromatin-remodeling complexes use ATP as an energy source to mobilize, eject, and exchange histones. The SWI/SNF complex alters nucleosome positioning and structure by sliding and ejecting nucleosomes to make the DNA more accessible to transcription factors and other chromatin regulators [65]. Noncoding RNAs (ncRNAs), including circular RNA (circRNA), long non-coding RNA (lncRNA), and microRNA (miRNA), can regulate other epigenetic processes, and be regulated by them. Some specific lncRNAs are extensively associated with chromatin remodeling and modification complexes, and target them to specific genomic loci to alter DNA methylation and the structure and modification of chromosomes [66].

Histone modifications affect chromatin structure, which plays an important role in gene regulation and carcinogenesis [67]. The most common changes observed in histone modification patterns in pancreatic cancer are methylation, acetylation, and ubiquitination. Multiple histone-modifying enzymes are involved in the dynamic regulation of these changes, and a balance between histone-modifying enzymes is critical for maintaining normal cellular function (Table 1). A common histone modification is methylation, which takes place on arginine, lysine, and histidine residues [68]. Lysine and arginine methyltransferases (KMTs and PRMTs) catalyze the transfer of the methyl group, while lysine demethylases (KDMs) control its removal. All methyltransferases use S-adenosyl methionine (SAM) as a substrate to transfer methyl groups onto the lysine and arginine residues of histones [69,70]. Histone acetylation is regulated by histone acetyltransferases (HATs) and histone deacetylases (HDACs). Acetylation can reduce the positive charge of lysine residues, leading to reduced binding between histone tails and negatively charged DNA, leaving the underlying DNA exposed [71]. Histone ubiquitination occurs primarily on H2A and H2B [72]. Ubiquitin (Ub) covalently binds to the target lysine residue under the continuous action of three proteins, the E1 activating enzyme, E2 conjugating enzyme, and E3 ligase [73]. E3 is essential and specifically required for histone ubiquitination [74]. Conversely, the deubiquitinating enzyme (DUB) is responsible for removing Ub [75].

## 4. Histone Modifications in Pancreatic Cancer EMT

### 4.1. Histone Methylation in EMT

#### 4.1.1. Histone Methylases

**PRMT1 in EMT**: Protein arginine methyltransferase 1 (PRMT1) is the principal methyltransferase that catalyzes the methylation of H4R3 and functions as a transcription co-activator [147]. PRMT1 expression is increased in patients with pancreatic cancer, and higher expression is associated with poorer prognosis [77,79]. PRMT1 can bind to the β-catenin promoter region, increasing the expression of the β-catenin protein in pancreatic cancer cells [79]. Stimulation of the Wnt/β-catenin signaling pathway promotes pancreatic cancer proliferation, migration, and invasion while regulating therapeutic resistance [148,149,150,151]. Increased levels of β-catenin in the nucleus can promote EMT-related gene expression. Furthermore, β-catenin can form a complex with E-cadherin to regulate the intercellular connections that are important in cell–cell adhesions [25,152].

The Hh signaling pathway is aberrantly activated during tumorigenesis in the pancreas [153]. Hh signaling induces the expression of EMT-related genes, such as PTCH1, WNT, and SNAI1, by activating Gli transcription factors. The methylation of Gli1 by PRMT1 at R597 improves the capacity of Gli1 to bind to its target gene promoter, thereby enhancing the transcriptional activity of Gli1 [81]. Moreover, ZEB1, a zinc finger E-box binding homeobox transcription factor, is crucial for the EMT process and abnormal expression of ZEB1 has been reported in pancreatic cancer [128]. Overexpression of ZEB1 can significantly reverse the anti-tumor effects induced by PRMT1 downregulation [77].

**PRMT5 in EMT**: PRMT5 catalyzes the methylation of arginine on H2AR3, H4R3, and H3R8, and acts primarily as a tumor-promoting factor [147,154]. Patients with pancreatic cancer who display higher levels of PRMT5 expression present lower overall survival [83,84]. PRMT5 induces the phosphorylation of the epidermal growth factor receptor (EGFR), upregulates the expression of β-catenin through the EGFR/AKT/β-catenin pathway, and promotes the EMT of pancreatic cancer cells to enhance tumor migration and invasion [83]. EGFR belongs to the membrane-bound ErbB/HER family of receptor tyrosine kinases (RTKs) and plays an important role in the maintenance of epithelial tissues. When EGFR signaling is altered, it becomes the grand orchestrator of epithelial transformation and promotes EGF-induced EMT in pancreatic cancer [155].

**SETD2 in EMT**: SETD2 is the sole enzyme responsible for H3K36me3 [156]. SETD2 is downregulated in pancreatic cancer, and a low expression of SETD2 is linked to poor clinical prognosis [93,94]. During pancreatic carcinogenesis, loss of SETD2 promotes KRAS-driven acinar-to-ductal metaplasia, as well as EMT and metastasis through sustained AKT activation and loss of α-catenin [94].

**KMT5 family in EMT**: KMT5A (SET8/PR-SET7) and KMT5B/C (SUV4-20H1/2) belong to the KMT5 family of lysine methyltransferases. H4K20 is monomethylated by KMT5A, before stepwise methylation occurs through the function of KMT5B/C from H4K20me1 to H4K20me2/me3 [21,157]. KMT5A upregulates the expression of stemness and EMT-related genes in pancreatic cancer cells by inducing the expression of the receptor tyrosine kinase-like orphan receptor 1 (ROR1) [99]. ROR1 is a transmembrane protein that induces EMT in breast cancer by enhancing EGFR signaling [158,159]. KMT5C has no direct effect on EMT-related transcription factors but instead regulates EMT by regulating the expression of MET-related transcription factors such as FOXA1, OVOL1, and OVOL2. Targeting KMT5C can activate epithelial transcription programs and inhibit the invasion and migration of pancreatic cancer cells [101].

**EZH2 in EMT**: Enhancer of zeste homolog 2 (EZH2) is the enzymatic subunit of the polycomb repressor complex 2 (PRC2) that promotes transcriptional silencing by methylating H3K27 [160]. EZH2 is overexpressed in patients with pancreatic cancer and is associated with poorer clinical outcomes [104,106]. The best characterized pathway involved in the induction of EMT is mediated by TGF-β signaling. TGF-β stimulates the expression of the SRY-box transcription factor 4 (SOX4) protein, which is an important developmental transcription factor involved in the regulation of the TGF-β signal transduction pathway [161]. SOX4 not only regulates the expression of EMT-related genes but also reprograms the epigenome to induce EMT by inducing EZH2 expression [162]. In pancreatic cancer, increased expression of SOX4 and EZH2 is associated with poor patient prognosis [116]. The downregulation of E-cadherin is a crucial step in EMT, which promotes metastasis by boosting cancer cell invasion and dissociation. EZH2 can stimulate the migration and invasion of pancreatic cancer by inhibiting the expression of E-cadherin [106]. EZH2 may achieve this through its interaction with the lncRNA MALAT-1, which promotes the proliferation and metastasis of pancreatic cancer [163].

#### 4.1.2. Histone Demethylases

**KDM2B in EMT**: KDM2B, also known as Ndy1, FBXL10, and JHDM1B, demethylates H3K36 and is overexpressed in human pancreatic cancer cells [164]. A core component of the Hippo signaling pathway is the adaptor protein MOB1 [165], which functions to depress Hippo transcriptional activity by increasing the phosphorylation of the yes-associated protein 1 (YAP) and tafazzin (TAZ) proteins. KDM2B transcriptionally inhibits the expression of MOB1 and promotes pancreatic cancer migration and invasion by promoting Hippo signaling [112]. KDM2B also inhibits the expression of several epithelial marker genes such as CDH1, miR200a, and CGN by modulating histone H1A ubiquitination [109].

**KDM3A in EMT**: KDM3A catalyzes the demethylation of H3K9me1/me2, which mediates transcriptional activation [166]. KDM3A mediates the upregulation of doublecortin-like kinase 1 (DCLK1), which is critical for the development and progression of pancreatic cancer during hypoxia. Under hypoxia, HIF1α activates KDM3A, which, in turn, increases DCLK1 mRNA expression [113]. DCLK1 activates EMT and promotes the migration and invasion of pancreatic cancer cells [167], while knockdown of DCLK1 reduces the expression of EMT transcription factors in these cells [168].

**KDM4B in EMT**: KDM4B is a member of the jumonji domain 2 (JMJD2) demethylase family [169], which catalyzes the demethylation of H3K9 and H3K36 [170]. KDM4B positively regulates the EMT process by activating the transcription of ZEB1, which plays a key role in TGF-β-induced EMT [116].

**KDM5A in EMT**: KDM5A, also known as JARID1A or RBP2, can remove the lysine of H3K4me2 and H3K4me3, resulting in activation or inhibition of transcription. KDM5A is overexpressed in pancreatic cancer cells, and is a member of the JMJC family of oxygen-sensitive enzymes [118,119,171]. KDM5A can be blocked by increased NOX4, which is the main endogenous source of reactive oxygen species (ROS). In pancreatic cancer cells under hypoxia, NOX4 expression is upregulated [172]. NOX4 regulates SNAIL1 transcription to induce EMT and promote the invasion and metastasis of pancreatic cancer cells [117].

**KDM6A in EMT**: KDM6A, also known as UTX, specifically catalyzes the demethylation of H3K27me3 [173] and is downregulated in pancreatic cancer cells [120,121,122]. The transcriptional activator hepatocyte nuclear factor-1a (HNF-1a) is thought to be a key tumor suppressor in pancreatic cancer [174], and recruits KDM6A to indirectly inhibit the expression of genes involved in tumorigenesis and EMT [122,123]. In pancreatic cancer cells, reductions in KDM6A lead to the increased expression of activin A, a member of the transforming growth factor-β superfamily of cytokines that activates a noncanonical p38 MAPK pathway to induce mesenchymal identity [120] and promote EMT [175,176].

### 4.2. Histone Acetylation in EMT

#### 4.2.1. Histone Acetylases

The CREB-binding protein (CBP) and p300 mediate histone lysine acetylation. Due to their considerable sequence homology and functional overlap, they are often referred to as a single entity (CBP/p300) [177]. CBP/p300 activates gene transcription through the acetylation of histone H3 lysine 27 (H3K27ac) [177]. Aberrant expression of p300 has been shown in pancreatic cancer cells [178]. p300 maintains the expression of the GATA-binding factor 6 (GATA6), thereby affecting the differentiation program regulated by this transcription factor [179]. GATA6 directly and indirectly inhibits dedifferentiation and EMT in pancreatic ductal adenocarcinoma (PDAC) [180]. P300/CBP-associated factor (PCAF) is a member of the GCN5-related N-acetyltransferase family that promotes transcription as a histone acetyltransferase [181]. PCAF/p300 has a synergistic effect in regulating the transcriptional repression/activation of ZEB1, forming the P300/PCAF/ZEB1 complex on the miR200c/141 promoter. PCAF/P300 induces the lysine acetylation of ZEB1 to activate miR200c transcription [182]. Loss of miR-200 family members and the upregulation of ZEB1 and ZEB2 are involved in the regulation of EMT. Transcriptional repression of the miR-200 family by ZEB1/ZEB2 is a major factor in the maintenance of mesenchymal characteristics and the induction of EMT [183]. Moreover, PCAF functions in the Ras ERK1/2 pathway and promotes the motility of pancreatic cancer cells, suggesting that PCAF is involved in pancreatic cancer EMT through multiple pathways [184].

#### 4.2.2. Histone Deacetylases

HDACs prevent gene expression. In pancreatic cancer, HDACs regulate proliferation, apoptosis, and metastasis [185]. High levels of HDAC1 and HDAC2 expression are linked to distant pancreatic cancer transitions, and both proteins enhance tumor invasiveness [129]. Pancreatic cancer cells proliferate and migrate more readily because of HDAC1 and HDAC2-mediated inhibition of E-cadherin expression in tumor cells [127,128]. HIF-1α plays a vital role as a transcriptional regulator of hypoxia-induced EMT, and a significant correlation exists between HIF-1α and HDAC1 expression in pancreatic cancer. In addition, high levels of these proteins are associated with a poorer prognosis for malignancies in patients, which may suggest that HDAC1 mediates HIF-1α stability through epigenetic regulation [186].

SIRT1 is a mammalian NAD+-dependent class III HDAC, which can regulate the Wnt/β-catenin signaling pathway by deacetylating β-catenin [187]. In the acinar-to-ductal metaplasia (ADM) of the pancreas, SIRT1 interferes with Wnt/β-catenin signaling by regulating the acetylation of β-catenin [133].

### 4.3. Histone Ubiquitination in EMT

#### 4.3.1. Histone Ubiquitinases

K119 is the most common ubiquitination checkpoint for H2A. The main E3 ubiquitin ligase of H2AK119ub1 is the catalytic subunit of the polycomb inhibitory complex (PRC1) [72]. This subunit consists of Ring1A and Ring1B and is regulated by Bmi1 (also known as PCGF4) [188]. The transcriptional repressor Snail can induce EMT by inhibiting the transcription of E-cadherin. In pancreatic cancer, Ring1A and Ring1B are critical for this regulatory process. Furthermore, EZH2 promotes Snail to recruit Ring1A and Ring1B more efficiently [189]. Concomitantly, Bmi-1 is overexpressed in pancreatic cancer, and this dysregulation is associated with abnormal expression of miRNAs. Overexpression of Bmi-1 can also promote EMT by downregulating E-cadherin [138,190].

#### 4.3.2. Histone Deubiquitinases

The ubiquitin-specific peptidases (USPs) are the primary members of the deubiquitinase family [191], where USP22 and USP28 are overexpressed in pancreatic cancer cells [139,140,141,142,143,144,145,146]. USP22 regulates the levels of H2Bub1 and H2Aub1 through deubiquitination, thereby promoting the transcription of downstream genes [192]. Focal adhesion kinase (FAK) is a cytoplasmic tyrosine kinase that mediates a variety of signal transduction pathways [193]. USP22 induces the occurrence of EMT in pancreatic cancer cells by activating FAK signaling. Upregulation of USP22 can increase the expression of the ZEB1 and Snail transcription factors, significantly reduce the expression of E-cadherin at cell–cell junctions, and upregulate the expression of mesenchymal markers [143].

USP28 regulates the process of histone H2A deubiquitination [194]. USP28 deubiquitinates and stabilizes the forkhead box M1 (FOXM1) transcription factor, which, in turn, activates the Wnt/β-catenin pathway and subsequently the expression of many EMT-related genes [146,195]. As a key mediator of Wnt/β-catenin signaling, FOXM1 mediates the nuclear accumulation of β-catenin and the ensuing downstream target gene expression in pancreatic cancer cells [196].

## 5. Histone Epigenetic Modulators in Pancreatic Cancer

Our understanding of the therapeutic potential of drugs targeting epigenetic modifiers has grown immensely in recent years. Histone modifiers are attractive targets for prospective therapy because they contain unique, druggable catalytic domains. Some FDA-approved inhibitors are already used in clinical practice, and others are currently in clinical trials [197]. Indeed, several drugs targeting EZH2 have been developed [198]. 3-Denitrogen-neplanocin A (DZNeP) and GSK126 are two newly discovered inhibitors of EZH2. Combination therapy of DZNeP and gemcitabine can act synergistically to inhibit pancreatic tumor cell growth and migration. DZNeP enhances the anti-proliferation activity of gemcitabine and significantly promotes apoptosis [199]. Similarly, GSK126 and BET bromine domain inhibitors have synergistic antitumor effects in the treatment of pancreatic cancer [200]. Thus, in combination with other therapeutics, EZH2 inhibitors possess great potential in the treatment of pancreatic cancer.

Histone deacetylase inhibitors (HDACIs) are becoming an important class of therapeutics in the treatment of pancreatic cancer. In preclinical studies, HDACIs have shown significant antitumor potential and low toxicity. HDACIs exert antitumor effects by stalling the cell cycle, activating apoptosis, inhibiting angiogenesis, and inhibiting metastasis [201]. However, when used alone, HDACIs are not as effective in the treatment of solid tumors. Many studies have now shown that HDACIs have synergistic effects with traditional cytotoxic and targeted therapeutic drugs against pancreatic cancer. The small molecule Domatinostat (4SC-202) is a class I selective HDACI that induces antitumor effects and sensitization to chemotherapy. It functions by reducing the expression of the transcription factor FOXM1, which impairs the redox homeostasis of cancer stem cells [202]. MPT0E028 is a novel pan-HDACI that targets both classes I and II HDACs. The co-administration of MPT0E028 and mitogen-activated protein kinase (MEK) inhibitors has synergistic effects in KRAS-mutated and KRAS-wild-type pancreatic cancer cells. This combination therapy induces a strong apoptotic response and significantly reduces cancer cell viability [203]. Additionally, the combination of radiation therapy with either CUDC-101 (simultaneously inhibits targets such as HDAC, EGFR, and HER2) or SAHA (inhibits class I and class II HDACs) could enhance radiation-induced cytotoxicity in human pancreatic cells [204].

Close interactions exist between the various histone epigenetic regulators, and many can be combined to synergistically regulate transcription. The use of dual inhibitors targeting different epigenetic regulators is an effective strategy to improve the safety and efficacy of single epigenetic target drugs and to overcome drug resistance. UDC-907 is a dual inhibitor of HDAC and PI3K that inactivates RAF-MEK-ERK signaling in pancreatic cancer cells by inhibiting the PI3K-AKT-mTOR pathway [205]. Metavert is a dual inhibitor of GSK3β and HDAC that induces cancer cell apoptosis and reduces both migration and the expression of stem cell markers, thereby slowing tumor and metastatic growth. Metavert has also been shown to provide synergistic effects with gemcitabine [206]. A new and effective strategy for the treatment of pancreatic cancer is 13A, a dual BET/HDAC inhibitor. 13A is more effective against pancreatic cancer than either BET inhibitors or HDAC inhibitors alone or when they are combined [207]. Finally, XP-524 is a dual-BET/EP300 inhibitor that prevents KRAS-induced tumor transformation. XP-524 can bind to anti-PD-1 antibodies, which reactivates the cytotoxic immune response [208].

## 6. Conclusions

The role of EMT in pancreatic cancer and the molecular mechanisms by which histone modifications regulate EMT-related molecules have been the subject of intense research over the past few years. The EMT process governs both physiological and pathological development, and abnormally active EMT phenotypes have been associated with pancreatic cancer initiation, progression, and therapeutic resistance. There is a now a large body of evidence demonstrating that many histone changes can reversibly control the expression of EMT markers during EMT and MET. Inhibitors of histone modification enzymes have been shown to raise the sensitivity of pancreatic cancer cells to chemotherapy and immunotherapy, thereby providing potential therapeutics. Combining histone modification enzyme inhibitors with established anti-tumor drugs is a promising treatment method. Thus, the elucidation of the molecular mechanisms by which histone-modifying enzymes regulate EMT in pancreatic cancers will enhance our understanding of the underlying tumorigenesis and metastatic progression, thereby facilitating the development of alternative treatments that may have important implications for disease outcomes.

## Figures and Tables

**Figure 1 ijms-24-04820-f001:**
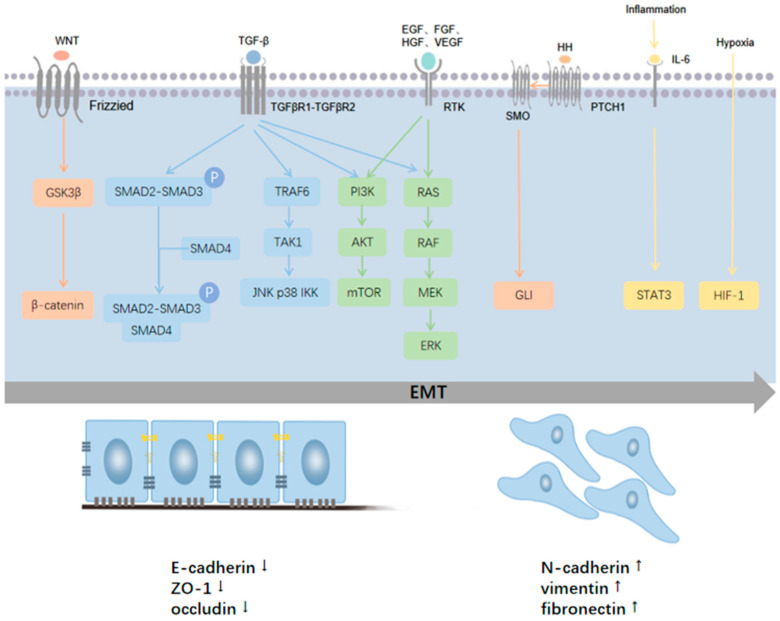
The signaling pathways associated with EMT in pancreatic cancer. TGF-β induces EMT through SMAD-mediated and non-SMAD signaling. TGF-β signals through a tetrameric complex of TβRI and TβRII to activate SMAD2 and SMAD3, which then combine with SMAD4. The trimeric SMAD complex regulates EMT-related transcriptional regulators. In addition, TGF-β can activate the ERK MAPK, Rho GTPase, and PI3K-AKT pathways. WNT signaling inhibits GSK3β to stabilize β-catenin, which translocates to the nucleus to promote the EMT-related gene expression program. In Hh signaling, Hh ligands bind to the PTCH1 to inhibit the SMO, activating the GLI transcription factors that induce the expression of EMT-related genes. Tyrosine kinase receptors promote EMT through the regulation of the PI3K/AKT and ERK/MAPK signaling pathways. The cell microenvironment also regulates EMT. During inflammation, IL-6 can promote EMT through the STAT3-mediated signaling pathway. Hypoxia in the tumor environment can promote EMT through HIF-1α, which interacts with the NF-κB transcriptional complex.

**Figure 2 ijms-24-04820-f002:**
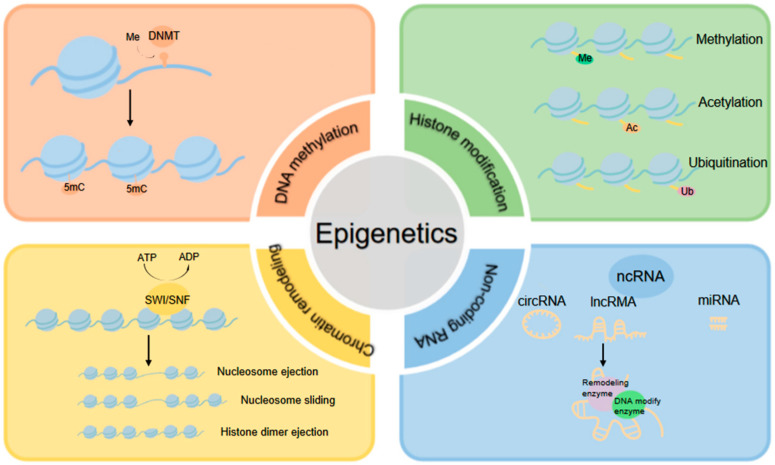
Epigenetic changes include DNA methylation, histone modification, chromatin remodeling, and changes to noncoding RNA profiles. DNA methylation involves the transfer a methyl group to the C5 position of cytosine in CpG dinucleotides to form 5-methylcytosine (5mC). Histone core protein possesses a characteristic tail that is enriched for lysine and arginine residues, which are subject to post-translational modifications. Acetylation, methylation, ribosylation, or phosphorylation of histone proteins can influence gene expression. The ncRNAs include circular RNA (circRNA), long non-coding RNA (lncRNA), and microRNA (miRNA). Some specific lncRNAs are associated with chromatin remodeling and modification complexes. Chromatin remodeling involves the ATP-dependent SWItch/sucrose non-fermentable (SWI/SNF) complex that functions in the mobilization, ejection, and exchange of histones.

**Table 1 ijms-24-04820-t001:** Histone-modifying enzymes in pancreatic cancer.

Epigenetic Mechanism	Family	Name	Synonyms	Site	Function
Histone methylation	PRMTs	PRMT1	HRMT1L2, HMT2, ANM1	H4R3	Promotes tumor initiation, proliferation, migration, and invasion; inhibition of PRMT1 reduces the resistance to gemcitabine and anti-PD-L1 checkpoint inhibitors [76,77,78,79,80,81,82]
		PRMT5	HRMT1L5, SKB1, HSL7	H3R2	Promotes tumorigenesis and aerobic glycolysis; induces EMT and promotes tumor migration and invasion; as a potential therapeutic target, has synergistic cytotoxicity with gemcitabine [83,84,85,86]
	KMTs	KMT1C	G9a, EHMT2	H3K9	Promotes tumor initiation and proliferation; inhibition of G9a reduces the resistance to gemcitabine [87,88,89,90]
		KMT2A	MLL1	H3K4	Upgrades the expression of programmed death-ligand 1 (PD-L1) in pancreatic cancer cells [91]
		KMT2D	MLL4	H3K4	Regulates tumor proliferation and metabolism [92]
		KMT3A	SETD2	H3K36	Promotes EMT and tumor migration [93,94]
		KMT3C/E	SMYD2/3	H4K5	Promotes tumor proliferation, migration, and invasion [95,96,97]
		KMT4	DOT1L	H3K79	Inhibits tumor apoptosis [98]
		KMT5A	SETD8	H4K20	Promotes tumor proliferation; enhances cancer cell stemness and induces EMT [99,100]
		KMT5C	SUV420H2	H4K20	Activates EMT, reduces cell stemness, and increases drug sensitivity when downregulated [101]
		KMT6	EZH2	H3K27	Promotes tumor proliferation, migration, and invasion; depletion of EZH2 sensitizes cancer cells to doxorubicin and gemcitabine [102,103,104,105,106]
	KDM	KDM1A	LSD1, AOF2	H3K4	Regulates tumor metabolism and promotes tumor proliferation [107,108]
		KDM2B	FBXL10, JHDM1B	H3K4, H3K36	Induces EMT and promotes tumor proliferation, migration, and invasion [109,110,111,112]
		KDM3A	JMJD1A, JHDM2A	H3K9	Regulates the immunological environment; promotes tumorigenesis and tumor metastasis and progression [108,113,114]
		KDM4A	JMJD2A	H3K36, H3K9	Impairs homologous recombination (HR), affecting DNA repair [115]
		KDM4B	JMJD2B	H3K9	Regulates EMT and promotes tumor metastasis and progression [116]
		KDM5A	JARID1A, RBP2	H3K4	Regulates tumor metabolism and EMT, promotes tumor migration and invasion [117,118,119]
		KDM6A	UTX	H3K27	Activates EMT and promotes tumor metastasis and progression [120,121,122,123]
Histone acetylation	KAT	p300/CBP		H3K18, H3K27	Regulates EMT; inhibition of p300 enhances the cytotoxicity of gemcitabine [124]
		HAT1		H4K5, H4K12	Promotes tumor progression; enhances the expression of PD-L1 in pancreatic cancer cells; sensitizes cancer cells to gemcitabine [125,126]
	KDAC	HDAC 1			Promotes EMT and tumor metastasis and invasion [127,128,129].
		HDAC 2			Inhibition of EZH2 sensitizes cancer cells to apoptosis [130,131]
		HDAC5			Inhibits the expression of PD-L1 in pancreatic cancer cells [132]
		SIRT1		H3K9, H3K14, H3K56, H4K16, H1K26	Promotes tumor initiation and EMT; inhibition of SIRT1 reduces the resistance to gemcitabine [133,134,135]
		SIRT6		H3K9, H3K56	Inactivation of SIRT6 promotes tumor metastasis and progression [136]
Histone ubiquitylation		Ring1A, Ring1B, Bmi1		H2AK119	Regulates EMT and cell cycle and promotes tumor invasion [137,138]
	USP	USP22		H2A, H2B	Regulates EMT and promotes tumor proliferation; downregulation of USP22 inhibits autophagy, promotes apoptosis, and regulates the immune microenvironment, which reduces the resistance to chemotherapy and immunotherapy [139,140,141,142,143,144,145]
		USP28		H2A	Promotes EMT and accelerates cancer cell growth [146]

## Data Availability

Not applicable.

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
