# Peer review of "Histone Modifications Represent a Key Epigenetic Feature of Epithelial-to-Mesenchyme Transition in Pancreatic Cancer"

_ijms, 2023, doi:10.3390/ijms24054820_

Round 1
Reviewer 1 Report
The manuscript entitled: “Epigenetic histone modification of epithelial-mesenchymal transition in pancreatic cancer: by Xu and Zhu concerns the role of histone modifications in EMT regulation in pancreatic cancer. I find the subject of the review of interest as pancreatic cancer is one of the deadliest diseases and therapeutic possibilities are very limited. Histone modifying enzymes are potential therapeutic targets and the portfolio of inhibitors t be used as single drug or in combination is continuously developed. Nevertheless, while reading this manuscript some major issues appeared that should be addressed by the authors before publication.
The title is a bit repetition of terms as histone modification is an epigenetic mechanism. It should be rather “histone modification as epigenetic mechanism driving epithelial-mesenchymal transition in pancreatic cancer” or similar… In general, major English editing is required.
Concerning figures:
Figure 1 seem to be more general and correspond to many types of cancer, not only pancreatic. In general, the authors should pay more attention in their statements to distinguish between the processes that are general to EMT and those which were precisely confirmed/discovered in pancreatic cancer.
Figure 2 is very poor. It does not explain the mechanism how DNA methylation or ncRNA work? We cannot see any differences between different histone modifications – at least different colors for acetylation, methylation or ubiquitination would do. The authors included in the figure and in the further text nucleosome remodeling as epigenetic mechanism, which was not mentioned previously in the Introduction part.
Tables in general need much attention.
Table 1 – the column with the names of the enzyme should have a name in the first line. In general this column should be wilder to accommodate multiple manes of the protein, or the authors should use just one, currently used name.
Table 2 – it contains both methylases and demethylases, therefore it should be stated in the table title.
Table 3 – similar as table 2,it contains both acetylases and deacetylases and the title should be changed adequately, besides HATs and HDAC class IV are included twice and HDACs are divided into classes and not types.
Why did the authors not include the table for ubiquitinating/deubiquitinating enzymes?
Other comments:
Line 46-47 – the regulation by non-coding RNA is not only related to posttranscriptional processes, it can play a role also during transcription.
Line 53 – some examples of enzymes regulating EMT would be desired.
Line 64 – remove “as”.
Line 67-79 this sentence is not clear to me.
Line 72 – English editing needed.
Line 81 – “So that the epithelial genes including E-cadherin, ZO- 1, and claudin were expressed at lower levels in cells undergoing EMT, whereas mesenchymal genes, such as N-cadherin, vimentin, and fibronectin, were expressed at higher levels[8]” – please re-write the sentence
Line 87 - what does it mean target-regulated.
Line 100 – please explain why CSC important in pancreatic cancer.
Line 108-120 – the whole paragraph is a mixture of surprising statements. It needs a main plot.
Line 127 – gene silencing.
Line 128 - better definition of chromatin needed.
Line 131 – the nucleosome contains two molecules of each histone protein, not histones. Editing needed.
Line 146 – there are many examples like that – not elegant sentence about correlation between gene/protein expression and patient survival.
Line 155-156 – difficult to read.
Line 157 – I assume authors meant the changes in those modifications.
Line 162 – please give examples of important enzymes or change the order of the paragraphs putting ATP-depended chromatin remodelers before histone modifying enzymes.
Line 172 – is this correct abbreviation (ncBAF?)
Line 175-176 - English editing needed.
Paragraph 4.1 – it is very difficult to read as different enzymes are mentioned in different parts of the text. It should be more structured.
Paragraphs about enzymes – difficult to understand as the authors present enzyme, than change the subject to some process and at the end explain the role of the enzyme in this process. I find it difficult to follow.
EZH2- do not repeat several times that it is overexpressed in pancreatic cancer.
Line 275 – “increases”.
Line 301 - English editing needed.
Line 305 – please explain what PDAC cells is.
Line 309 – English editing needed. (Histone acetylation does not catalase, it is catalyzed by specific enzymes).
Line 316 - English editing needed.
Line 327 – explain HDACis.
Line 339 – Please, re-write the sentence.
Line 343 – should be one sentence.
Overall, considering the conclusions, a paragraph about HDACi in preclinical/clinical studies is a bit missing.
Author Response
Response to Reviewer 1 Points
Point 1: The title needs to be corrected. The title is a bit repetition of terms as histone modification is an epigenetic mechanism. It should be rather “histone modification as epigenetic mechanism driving epithelial-mesenchymal transition in pancreatic cancer” or similar... In general, major English editing is required.
Response 1: Thank you for your sincere suggestion. We accept your suggestion to change the title to "Histone Modifications represent a Key Epigenetic Feature of Epithelial-to-Mesenchyme Transition in Pancreatic Cancer". And we have carefully and thoroughly proofread the manuscript to correct all the grammar and typos.
Point 2: Figure 1 seem to be more general and correspond to many types of cancer, not only pancreatic. In general, the authors should pay more attention in their statements to distinguish between the processes that are general to EMT and those which were precisely confirmed/discovered in pancreatic cancer.
Response 2: In Figure 1, we furtherly describe the molecular mechanism of the EMT process in detail. For your suggestion of "distinguishing the general process of EMT from the process of precise identification/discovery in pancreatic cancer", we state below the relevant mechanisms involved in EMT affecting the process of pancreatic cancer occurrence, development, and treatment resistance.
Point 3: Figure 2 is very poor. It does not explain the mechanism how DNA methylation or ncRNA work? We cannot see any differences between different histone modifications – at least different colors for acetylation, methylation or ubiquitination would do. The authors included in the figure and in the further text nucleosome remodeling as epigenetic mechanism, which was not mentioned previously in the Introduction part.
Response 3: We have supplemented this as follows “DNA methylation transfers a methyl group to the C5 position of cytosine in CpG dinucleotides to form 5-methylcytosine (5mC). The DNA (cytosine-5)-methyltransferase 3A (DNMT3A) and DNMT3B enzymes are responsible for de novo DNA methylation, which is maintained by DNMT1. Chromatin-remodeling complexes use ATP as an energy source to mobilize, eject, and exchange histones. The SWI/SNF complex alters nucleosome positioning and structure by sliding and ejecting nucleosomes to make the DNA more accessible to transcription factors and other chromatin regulators. Noncoding RNAs (ncRNAs), including circular RNA (circRNA), long non-coding RNA (lncRNA), and microRNA (miRNA), can regulate other epigenetic processes, and be regulated by them. Some specific lncRNAs are extensively associated with chromatin remodeling and modification complexes, and target them to specific genomic loci to alter DNA methylation and the structure and modification of chromosomes. ”
Point 4: Table 1 – the column with the names of the enzyme should have a name in the first line. In general this column should be wilder to accommodate multiple manes of the protein, or the authors should use just one, currently used name.
Response 4: We have supplemented “name” and “Synonyms” in the table 1.
Point 5: Table 2 – it contains both methylases and demethylases, therefore it should be stated in the table title. Table 3 – similar as table 2,it contains both acetylases and deacetylases and the title should be changed adequately, besides HATs and HDAC class IV are included twice and HDACs are divided into classes and not types. Why did the authors not include the table for ubiquitinating/deubiquitinating enzymes?
Response 5: We have removed this section to better describe the central content.
Point 6: Line 46-47 – the regulation by non-coding RNA is not only related to posttranscriptional processes, it can play a role also during transcription.
Response 6: We revise this concept to “The major epigenetic regulators include DNA methylation, histone modification, chromatin remodeling, and non-coding RNAs.”
Point 7: Line 53 – some examples of enzymes regulating EMT would be desired.
Response 7: This will be the main statement below, so we will not describe it in detail here.
Point 8: Line 64 – remove “as”.
Response 8: We revise this concept to “EMT is induced by a variety of extracellular stimuli and intracellular signaling pathways, including the transforming growth factor (TGF), Wnt, and hedgehog (Hh) signaling pathways, etc. ”
Point 9: Line 67-79 this sentence is not clear to me.
Response 9: We have restated the signaling pathways involved in EMT.
Point 10: Line 72 – English editing needed.
Response 10: We revise this concept to “In the absence of Wnt signaling, β-catenin is phosphorylated by casein Kinase 1 (CK1) and adenomatous polyposis coli (APC)/AXIN/glycogen synthase kinase-3 beta (GSK3β) complexes, prior to being ubiquitinated and degraded by proteasomes.”
Point 11: Line 81 – “So that the epithelial genes including E-cadherin, ZO- 1, and claudin were expressed at lower levels in cells undergoing EMT, whereas mesenchymal genes, such as N-cadherin, vimentin, and fibronectin, were expressed at higher levels[8]” – please re-write the sentence.
Response 11: We revise this concept to “The activation of EMT leads to a downregulation in the expression of epithelial genes, including E-cadherin, ZO-1, and occludin. Concurrently, cells acquire a mesenchymal morphology and express mesenchymal markers, such as N-cadherin, vimentin, and fibronectin.”
Point 12: Line 87 - what does it mean target-regulated.
Response 12: Considering the length of the article, We have removed these sentences.
Point 13: Line 100–please explain why CSC important in pancreatic cancer.
Response 13: We revise this concept to “Cancer stem cells (CSCs) are undifferentiated quiescent cells that possess properties of self-renewal and cellular plasticity. Pancreatic cancer stem cells (PCSCs) are subsets of pancreatic cancers with specific cell surface markers that play important functions during tumor recurrence and therapeutic resistance.”
Point 14: Line 108-120–the whole paragraph is a mixture of surprising statements. It needs a main plot.
Response 14: We have reorganized this paragraph to primarily state how inflammation and the hypoxic microenvironment regulate EMT in pancreatic cancer.
Point 15: Line 127–gene silencing.
Response 15: We revise this concept to “Epigenetic modifications are reversible changes that regulate gene expression without altering DNA sequences.”
Point 16: Line 128 - better definition of chromatin needed.
Response 16: We revise this concept to “Chromatin is the carrier of genetic information, and the nucleosome is its basic unit.”
Point 17: Line 131–the nucleosome contains two molecules of each histone protein, not histones. Editing needed.
Response 17: We revise this concept to “Each nucleosome consists of a histone octamer composed of the core histones (two copies each of H2A, H2B, H3, and H4), which is wrapped in approximately 147 bp of DNA.”
Point 18: Line 146–there are many examples like that–not elegant sentence about correlation between gene/protein expression and patient survival.
Response 18: We have removed this section to better describe histone modifications.
Point 19: Line 155-156–difficult to read.
Response 19: We revise this concept to “Histone modifications affect chromatin structure, which plays an important role in gene regulation and carcinogenesis .”
Point 20: Line 157–I assume authors meant the changes in those modifications.
Response 20: We revise this concept to “The most common changes observed in histone modification patterns in pancreatic cancer are methylation, acetylation, and ubiquitination.”
Point 21: Line 162–please give examples of important enzymes or change the order of the paragraphs putting ATP-depended chromatin remodelers before histone modifying enzymes.
Response 21: We have removed this section to better describe histone modifications.
Point 22: Line 172–is this correct abbreviation (ncBAF?)
Response 22: We have removed this section to better describe histone modifications.
Point 23: Line 175-176 - English editing needed.
Response 23: We have removed this section to better describe histone modifications.
Point 24: Paragraph 4.1–it is very difficult to read as different enzymes are mentioned in different parts of the text. It should be more structured.
Response 24: We have summarized the content of enzymes.
Point 25: Paragraphs about enzymes–difficult to understand as the authors present enzyme, than change the subject to some process and at the end explain the role of the enzyme in this process. I find it difficult to follow.
Response 25: We have summarized the content of enzymes.
Point 26: EZH2- do not repeat several times that it is overexpressed in pancreatic cancer.
Response 26: We have removed the duplicate.
Point 27: Line 275–”increases”.
Response 27: We have corrected it to “A core component of the Hippo signaling pathway is the adaptor protein MOB1, which functions to depress Hippo transcriptional activity by increasing the phosphorylation of the yes-associated protein 1 (YAP) and tafazzin (TAZ) proteins.”
Point 28: Line 301 - English editing needed.
Response 28: We revise this concept to “KDM6A, also known as UTX, specifically catalyzes the demethylation of H3K27me3 and is down-regulated in pancreatic cancer cells. ”
Point 29: Line 305–please explain what PDAC cells is.
Response 29: PDAC cells are pancreatic ductal adenocarcinoma cells. We revise this word to “ pancreatic cancer cells”.
Point 30: Line 309–English editing needed. (Histone acetylation does not catalase, it is catalyzed by specific enzymes).
Response 30: We revise this concept to “Histone acetylation is regulated by histone acetyltransferases (HATs) and histone deacetylases (HDACs). Acetylation can reduce the positive charge of lysine residues, leading to reduced binding between histone tails and negatively charged DNA, leaving the underlying DNA exposed. ”
Point 31: Line 316 - English editing needed.
Response 31: We have carefully and thoroughly proofread the manuscript to correct all the grammar and typos.
Point 32: Line 327–explain HDACis.
Response 32: We explain HDACis in the newly added part about "Histone epigenetic modulators in pancreatic cancer" at the end of the article.
Point 33: Line 339–Please, re-write the sentence.
Response 33: We revise this concept to “P300/CBP-associated factor (PCAF) is a member of the GCN5-related N-acetyltransferase family that promotes transcription as a histone acetyltransferase .”
Point 34: Line 343–should be one sentence.
Response 34: We revise this concept to “ Moreover, PCAF functions in the Ras ERK1/2 pathway and promotes the motility of pancreatic cancer cells, suggesting that PCAF is involved in pancreatic cancer EMT through multiple pathways.”
Point 35: Overall, considering the conclusions, a paragraph about HDACi in preclinical/clinical studies is a bit missing.
Response 35: Response 36: As you said above "Histone modifying enzymes are potential therapeutic targets and the portfolio of inhibitors t be used as single drug or in combination is continuously developed." We have added a part about "Histone epigenetic modulators in pancreatic cancer" at the end of the article, which includes more statements on HDACi.
Reviewer 2 Report
The authors have attempted to review the literature related to epigenetic regulation of EMT in pancreatic cancer. Even though the central scope of this review is interesting, the authors do not manage to get their message across, as they offer too much background information in a not so well-organized manner, while the specific information about the regulation of EMT by histone modifications in pancreatic cancer is kind of “buried” in the text.
The authors need to gather all background information in one section, organize it better and limit the number of references. I’ll give one example: E-cadherin is first mentioned in line 81 as an epithelial gene that is downregulated in EMT. Then again in Line 264 the authors repeat this and also add new refs. The latter is not needed.
Furthermore, when they describe research work about pancreatic cancer, this is done in a very generic way, resulting in the reader not acquiring any useful knowledge on the topic. For example, in lines 88-89 the authors write “Zeb1 can help suppress ZO-1 and claudin-1 expression, which is crucial for the pancreatic cancer EMT process”, however, they do not explain why this is the case, how this was shown, in what model system etc. Other example, in lines 151-152: “… whereas DNMT3A and DNMT3B can be employed as possible therapeutic targets[78, 79]. DNMT3A can control the ability of tumor cells to respond to chemotherapy [80]” . These are very general statements with no explanation how these results were obtained, in what system, what kind of chemotherapy etc
And so on and so forth…
Moreover, information that is not important for the comprehension of the work described in this review should be omitted. For example, there is no reason to describe all families of the epigenetic enzymes mentioned in this review and table 2 is completely unnecessary.
The figures are too simple and depict very well-known notions in the field offering no useful information to the reader. They should be replaced, ideally, by figures that are related with the central scope of this review.
There are several spelling mistakes, incorrect usage of tenses and of singular/plural in verbs etc. Language editing is needed.
On top of the above, what I find most troubling is the fact that the authors fail to give correct definitions for basic concepts in the field, such as for the term “epigenetics” or “chromatin” (see below), which may be due to carelessness or incorrect use of the English language, however it does result to fallacies.
Specific Points
The title needs to be corrected.
In line 35, after the physiological roles of EMT are mentioned, its relevance to cancer in general should be described first before discussing specifically pancreatic cancer.
Line 44: “Epigenetics is the regulatory process of modifying DNA or histones post-translationally without altering the DNA sequence itself”
This definition is inaccurate in multiple aspects. Please correct it.
Lines 45-46: “… include DNA methylation, chromatin modification, post-translational gene regulation of non-coding RNAs, etc”
This is a problematic sentence. DNA methylation is included in chromatin modifications, the authors probably mean post-translational histone modifications. Also, non-coding RNAs are involved in post-transcriptional regulation of gene expression.
Lines 47-49: “These modifications can alter the accessibility of the
promoter and the structure of the chromatin, which results in aberrant gene expression and aids in the initiation, proliferation, and metastasis of tumors …”
In the previous sentence the authors mention ncRNAs, which are not modifications and their mode of action is different. Need to rephrase this sentence for example they could say “these mechanisms…”.
In addition, the second part of this sentence is over-simplistic as not all aberrant gene expression programs lead to tumor initiation. Need to rephrase.
Line 62: Please give better definition of EMT and its relation to cancer.
Line 117: “among which HIF-1α is the hypoxic response element in the promoter region of the EMT-TF gene”
This is incorrect. Please correct it.
Line 126: “Epigenetics is a reversible pathway that regulates gene expression” see above for correcting definition of epigenetics. What the authors probably refer to is “epigenetic mechanisms”.
Line 128: “Chromatin is where eukaryotic organisms store their
genetic material.”
Line 130: “The nucleosome is then wound into a hollow spiral tube to become
chromatin.”
Please correct the above sentences, as they are inaccurate.
Line 131: “ Histones H2A, H2B, H3, and H4 each contain two molecules…”
Please rephrase so as to explain correctly the structure of nucleosome.
Lines 134-5: “These alterations may change the accessibility of the promoter and the structure of the chromatin, which will lead to aberrant gene expression”
These lines are almost identical to lines 47-49.
Finally, I would strongly recommend a) to take out all the information about epigenetic mechanisms/enzymes in pancreatic cancer that are not related to EMT. This will give room to the authors to expand more on the central topic of this review that is histone modifications regulating EMT in pancreatic cancer and b) to also omit the parts about DNA methylation and ATP-dependent chromatin remodeling in order to keep up with the title of the review.
Author Response
Response to Reviewer 2 Points
Point 1: The title needs to be corrected.
Response 1: Thanks for your sincere suggestion. We will change the title to “Histone Modifications represent a Key Epigenetic Feature of Epithelial-to-Mesenchyme Transition in Pancreatic Cancer ”
Point 2: In line 35, after the physiological roles of EMT are mentioned, its relevance to cancer in general should be described first before discussing specifically pancreatic cancer.
Response 2: We have added “EMT enables cancer cells at the primary site to acquire motility and invasiveness, which can drive the malignant progression of tumors. Furthermore, EMT also can regulate a variety of cancer features, such as tumor cell stemness, adaptation to microenvironmental alterations, and resistance to therapy.” between these two sentences.
Point 3: Line 44:”Epigenetics is the regulatory process of modifying DNA or histones post-translationally without altering the DNA sequence itself”. This definition is inaccurate in multiple aspects. Please correct it.
Response 3: We revise this concept to “Epigenetics describe heritable changes in cellular phenotypes that are independent of alterations in the DNA sequence. ”
Point 4: Lines 45-46: “… include DNA methylation, chromatin modification, post-translational gene regulation of non-coding RNAs, etc”. This is a problematic sentence. DNA methylation is included in chromatin modifications, the authors probably mean post-translational histone modifications. Also, non-coding RNAs are involved in post-transcriptional regulation of gene expression.
Response 4: We revise this sentence to “The major epigenetic regulators include DNA methylation, histone modification, chromatin remodeling, and non-coding RNAs. ”
Point 5: Lines 47-49: “These modifications can alter the accessibility of the promoter and the structure of the chromatin, which results in aberrant gene expression and aids in the initiation, proliferation, and metastasis of tumors …”. In the previous sentence the authors mention ncRNAs, which are not modifications and their mode of action is different. Need to rephrase this sentence for example they could say “these mechanisms…”.In addition, the second part of this sentence is over-simplistic as not all aberrant gene expression programs lead to tumor initiation. Need to rephrase.
Response 5: We revise these sentences to “Epigenetic changes are associated with aberrant gene functions and altered patterns of gene expression. The major epigenetic regulators include DNA methylation, histone modification, chromatin remodeling, and non-coding RNAs. These molecular regulatory mechanisms can alter promoter accessibility and chromatin structure, so as to affect cellular gene expression and influence tissue homeostasis. ”
Point 6: Line 62: Please give better definition of EMT and its relation to cancer.
Response 6: We revise these sentences to “EMT leads to significant phenotypic alterations in cancer cells that are associated with their invasion, spread, and metastasis. Epithelial cells maintain cell-cell contacts through tight junctions, adhesive junctions, desmosomes, and gap junctions”
Point 7: Line 117: “among which HIF-1α is the hypoxic response element in the promoter region of the EMT-TF gene”.This is incorrect. Please correct it.
Response 7: We revise these sentences to “The adaptive response of pancreatic cancer cells to hypoxia is mainly mediated by hypoxia-inducible factors (HIFs), which bestow more aggressive and therapeutically resistant phenotypes. During hypoxia, the ubiquitin-proteasome degradation pathway mediated by HIF-1 is inhibited. As a consequence, HIF-1α accumulates in cells and binds to HIF-1β to form a functional transcriptional complex. ”
Point 8: Line 126: “Epigenetics is a reversible pathway that regulates gene expression” see above for correcting definition of epigenetics. What the authors probably refer to is “epigenetic mechanisms”.
Response 8: We revise this sentence to “Epigenetic modifications are reversible changes that regulate gene expression without altering DNA sequences.”
Point 9: Line 128: “Chromatin is where eukaryotic organisms store their genetic material.” Line 130: “The nucleosome is then wound into a hollow spiral tube to become chromatin.” Please correct the above sentences, as they are inaccurate.Line 131: “ Histones H2A, H2B, H3, and H4 each contain two molecules…”. Please rephrase so as to explain correctly the structure of nucleosome.
Response 9: We revise these sentences to “Chromatin is the carrier of genetic information, and the nucleosome is its basic unit. Each nucleosome consists of a histone octamer composed of the core histones (two copies each of H2A, H2B, H3, and H4), which is wrapped in approximately 147 bp of DNA.”
Point 10: Lines 134-5: “These alterations may change the accessibility of the promoter and the structure of the chromatin, which will lead to aberrant gene expression”.These lines are almost identical to lines 47-49.
Response 10: We will delete this sentence.
Point 11: Finally, I would strongly recommend a) to take out all the information about epigenetic mechanisms/enzymes in pancreatic cancer that are not related to EMT. This will give room to the authors to expand more on the central topic of this review that is histone modifications regulating EMT in pancreatic cancer and b) to also omit the parts about DNA methylation and ATP-dependent chromatin remodeling in order to keep up with the title of the review.
Response 11: We will delete most of the information about epigenetic mechanisms/enzymes in pancreatic cancer that are not related to EMT and the parts about DNA methylation and ATP-dependent chromatin remodeling
Round 2
Reviewer 2 Report
The authors have adequately addressed all comments made by the reviewers and the manuscript has improved significantly.
Two minor things:
a) The figures should be moved after the first paragraph of each section and not at the beginning.
b) Figure legend 1 should briefly describe what we see in the figure. The title alone is not enough.
Author Response
Response to Reviewer 2 Points
Point 1: The figures should be moved after the first paragraph of each section and not at the beginning.
Response 1: Thanks for your sincere suggestion. We have moved The figures after the first paragraph of each section.
Point 2: Figure legend 1 should briefly describe what we see in the figure. The title alone is not enough.
Response 2: We have added “Figure 1. The signaling pathways associated with EMT in pancreatic cancer. TGF-β induces EMT through SMAD-mediated and non-SMAD signaling. TGF-β signals through a tetrameric complex of TβRI and TβRII to activate SMAD2 and SMAD3, which then combine with SMAD4. The trimeric SMAD complex regulates EMT-related transcriptional regulators. In addition, TGF-β can activate the ERK MAPK, Rho GTPase, and PI3K-AKT pathways. WNT signaling inhibits GSK3β to stabilize β-catenin, which translocates to the nucleus to promote the EMT-related gene expression program. In HH signaling, Hh ligands bind to the PTCH1 to inhibit the SMO, activating the GLI transcription factors that induce the expression of EMT-related genes. Tyrosine kinase receptors promote EMT through the regulation of the PI3K/AKT and ERK/MAPK signaling pathways. The cell microenvironment also regulates EMT. During inflammation, IL-6 can promote EMT through the STAT3-mediated signaling pathway. Hypoxia in the tumor environment can promote EMT through HIF-1α, which interactes with the NF-κB transcriptional complex.”